# Identification of *Francisella tularensis* Subspecies in a Clinical Setting Using MALDI-TOF MS: An In-House *Francisella* Library and Biomarkers

**DOI:** 10.3390/microorganisms11040905

**Published:** 2023-03-30

**Authors:** Maaike C. de Vries, B. J. A. Hoeve-Bakker, Maaike J. C. van den Beld, Amber C. A. Hendriks, Airien S. D. Harpal, Ramón C. E. A. Noomen, Frans A. G. Reubsaet

**Affiliations:** Centre for Infectious Disease Control, National Institute for Public Health and the Environment (RIVM), 3721 MA Bilthoven, The Netherlands; dieneke.hoeve@rivm.nl (B.J.A.H.-B.); maaike.van.den.beld@rivm.nl (M.J.C.v.d.B.); amber.hendriks@rivm.nl (A.C.A.H.); ramon.noomen@rivm.nl (R.C.E.A.N.);

**Keywords:** tularemia, MALDI-TOF MS, biomarkers, in-house library, *Francisella*, identification, protein identification

## Abstract

*Francisella tularensis* is a zoonotic bacterium that is endemic in large parts of the world. It is absent in the standard library of the most applied matrix-assisted laser desorption ionization-time of flight mass spectrometry (MALDI-TOF MS) systems: the Vitek MS and the Bruker Biotyper system. The additional Bruker MALDI Biotyper Security library contains *F. tularensis* without subspecies differentiation. The virulence of *F. tularensis* differs between the subspecies. The *F. tularensis* subspecies (ssp.) *tularensis* is highly pathogenic, whereas the subspecies *holarctica* displays lower virulence and subspecies *novicida* and *F. tularensis* ssp. *mediasiatica* are hardly virulent. To differentiate the *Francisellaceae* and the *F. tularensis*-subspecies, an in-house *Francisella* library was built with the Bruker Biotyper system and validated together with the existing Bruker databases. In addition, specific biomarkers were defined based on the main spectra of the *Francisella* strains supplemented with in silico genome data. Our in-house *Francisella* library accurately differentiates the *F. tularensis* subspecies and the other *Francisellaceae*. The biomarkers correctly differentiate the various species within the genus *Francisella* and the *F. tularensis* subspecies. These MALDI-TOF MS strategies can successfully be applied in a clinical laboratory setting as a fast and specific method to identify *F. tularensis* to subspecies level.

## 1. Introduction

*Francisella tularensis* is a Gram-negative zoonotic bacterium which is classified as a class 3 biological agent and category A bioterrorism agent by the United States Centers for Disease Control and Prevention (CDC, Atlanta, GA, USA) and as category I by the European Medicine Agency (EMA).

The natural life cycle of *F. tularensis* is concentrated on lagomorphs and rodents, whereas arthropods and mosquitos can play a role as vectors in transmission to humans [1]. After handling infected carcasses or following an infected arthropod bite, ulceroglandular tularemia may arise. Consumption of contaminated water or food may result in oropharyngeal tularemia. Inhaling contaminated aerosols can result in pneumonia. The most severe form of illness is tularemia sepsis. Due to the low inoculation dose and the ability to penetrate an intact skin, *F. tularensis* is a serious threat for laboratory-acquired infections [2].

*F. tularensis* can be divided into four subspecies, namely, *F. tularensis* subspecies (ssp.) *tularensis*, *F. tularensis* ssp. *holarctica*, *F. tularensis* ssp. *mediasiatica* [3], and *F. tularensis* ssp. *novicida* [4]. *F. tularensis* ssp. *tularensis*, which is also known as type A, is the most pathogenic subspecies and should be handled under biosafety level 3 (BSL-3) conditions. This subspecies is found in North America [5]. *F. tularensis* ssp. *holarctica*, which is also known as type B, displays lower virulence and can be handled under BSL-2 conditions. This subspecies is present in the Northern Hemisphere [5]. Both *F. tularensis* ssp. *mediasiatica*, found in Central Asia [3], and *F. tularensis* ssp. *novicida*, found in in the Northern Hemisphere and Australia, are less virulent [6]. No human cases have been described for *F. tularensis* ssp. *mediasiatica*, whereas infection with *F. tularensis* ssp. *novicida* is rare and mainly restricted to immunocompromised patients [6,7]. *F. tularensis* ssp. *novicida* is an outlier among the other *F. tularensis* subspecies [8], because it is transmitted from salt or brackish water and not found in animals [6]. Additionally, it differs genetically from the other *F. tularensis* subspecies [6,9,10,11].

Due to this wide range of clinical symptoms and virulence and the possibility of intentional dissemination, clinical laboratories need a fast and accurate method to identify *F. tularensis* and distinguish its subspecies. Matrix-assisted laser desorption ionization time-of-flight mass spectrometry (MALDI-TOF MS) could have this potential. The Bruker Biotyper MALDI-TOF MS system is increasingly used for the identification of bacteria in clinical laboratories. The Vitek MS (Biomerieux) is also available for this purpose. In the standard library of both the Vitek MS [12] and the Bruker Biotyper system (MALDI Biotyper library), *F. tularensis* is not included. The MALDI Biotyper library, however, can be supplemented with the MALDI Biotyper Security library (SR) for potential bioterrorism agents including *F. tularensis* but without *F. tularensis* subspecies differentiation. To overcome this, an in-house library was created to distinguish *F. tularensis* subspecies [13]. *Francisella*-specific biomarkers were studied for *F. tularensis*, with and without subspecies differentiation, and for *F. philomiragia*, *F. halioticida* and *F.salimarina* [13,14,15,16,17,18,19]. An in-house database was also described limited to *F. tularensis* ssp. *holarctica* and *novicida* and *F. philomiragia* [18] and one limited to different MLVA end PFGE-types of *F. tularensis* ssp. *holarctica* from Spain and the Czech Republic [16].

In this study, MALDI-TOF MS was validated for fast and accurate detection of the complete range of clinically relevant *Francisella* species including *F. tularensis* and differentiation to subspecies level within a clinical setting by building an in-house *Francisella* library and determining specific biomarkers.

## 2. Materials and Methods

Both main spectra (MSPs) and regular spectra were made of the clinically relevant *Francisellaceae*, including *F. tularensis* with the four different subspecies that can be encountered in routine clinical practice and of the closely related *Legionella pneumophila*. The in-house *Francisella*-specific database was built using the MSPs and was verified, together with the BDAL and SR database, using the regular spectra. The MSPs were also used to determine differentiating biomarkers (Figure 1).

### 2.1. Strains Used in This Study

All *F. tularensis* strains, except for the wild-type strains, were kindly provided by the repository of the European Union project for the “Establishment of Quality Assurances for Detection of Highly Pathogenic Bacteria of Potential Bioterrorism Risk” (EQADeBa; Grant number 2007204). Strains genetically or phenotypically related to *F. tularensis* were selected as negative control strains and were obtained from “Deutsche Sammlung von Mikroorganismen und Zellkulturen GmbH” (DSMZ, Braunschweig, Germany), “Culture Collection, University of Göteborg” (CCUG, Göteborg, Sweden), and “American Type Culture Collection” (ATCC, Manassas, VA, USA) (Table 1).

*F. halioticida* and *F.*[*Wolbachia*] *persica* were excluded because special culture media are required for growth [20] and therefore they are not likely to be encountered in a routine clinical laboratory. Not validly published species were also excluded.

### 2.2. Verification of Strain Identity

The identity of all strains was verified by 16S rDNA Sanger sequencing and real-time PCR targeting the outer membrane gene *tul4* [21] and the helicase gene (*FT-heli*) [22]. In addition, wild-type strains were analysed using conventional biochemical techniques and fatty acid analysis (Sherlock Microbial Identification System (MIDI Labs, Inc., Newark, NJ, USA) using the CLIN 6.1 and BTR 3.1 library) according to the polyphasic taxonomy [23].

A loop full of bacterial culture was suspended in 100 µL of sterile water and heat inactivated at 100 °C for 30 min under BSL-3 conditions. All further molecular techniques were applied at BSL-2 conditions. PCR amplification of the DNA consisted of 10 min pre-denaturation at 95 °C, 35 cycles of denaturation at 95 °C for 30 s, annealing at 54 °C for 30 s, and elongation at 72 °C for 80 s using the primers described in Table 2 [24]. Partial (nt8-575 accession J01695) 16S rDNA Sanger sequence analysis was performed on the amplified DNA with a 3130XL Genetic Analyzer (Applied Biosystems, Nieuwerkerk aan de IJssel, the Netherlands). The 16S rDNA sequences were analysed using Clone Manager 9.3 (Sci-Ed Software, Cary, NC, USA). The sequence of the 16S rRNA gene of every strain was compared to the sequence of their type strain. The 16S rDNA sequence of the type strains as determined in our laboratory is compared to available 16S rDNA sequences of those strains in GenBank (National Center for Biotechnology Information (NCBI), Bethesda, USA). The in-house sequence of the type strains was compared to GenBank acessionnr. AY968223 (*F. tularensis* ssp. *tularensis*), AY968231 (*F. tularensis* ssp. *holarctica*), AY968237 (*F. tularensis* ssp. *novicida*), AJ698863 (*F. tularensis* ssp. *mediasiatica*), FN252413 (*F. hispaniensis*), DQ295795 (*F. noatunensis* ssp. *noatunensis*), EU683030 (*F. orientalis*), AY968239 (*F. philomiragia*), FJ591095 (*A. guangzhouensis*), M59157 (*L. pneumophila*). The 16S rDNA sequences of all *F. tularensis* ssp. *tularensis* were compared to *F. tularensis* ssp. *tularensis* strain FSC 053 (AY968223) because only short stretches of the type strain sequence are deposited in GenBank [25]. The type strain and the 16S rDNA sequence of *F. tularensis* ssp. *holarctica* are not available in any known culture collection [26] or database, respectively. Strain *F. tularensis* ssp. *holarctica* FSC257 (AY968231) was selected for comparison of the 16S rDNA sequence.

The *F. tularensis* gene *tul4* [21] was targeted using a fluorescence resonance energy transfer (FRET)-based real-time PCR. *FT-heli* [22] was chosen to differentiate *F. tularensis* ssp. *holarctica* from the other subspecies. A PCR product of each target was cloned into the CR2.1-TOPO plasmid (TOPO TA Cloning Kit, Invitrogen, Breda, Netherlands) to serve as internal control. The quality of the cloned fragments was confirmed by sequence analysis. The LightCycler FastStart DNA Master^plus^ HybProbe kit (Roche Diagnostics GmbH, Berlin, Germany) was used with a reaction mixture containing 10 pmoles of both forward and reverse primer and 4 pmoles of the fluorescein-labelled probe and the CAL Fluor Red 635-labelled probe each (Table 2). Four µL DNA of the described strains (Table 1), plasmid solution, or PCR-grade water (in case of negative controls) was added to a final volume of 20 μL. All samples were run with and without plasmid control (10 fg per reaction) to check for inhibition.

The PCR was performed on the LightCycler 480 system (Roche Diagnostics GmbH, Berlin, Germany) and consists of 10 min pre-denaturation at 95 °C; 45 cycles of denaturation at 95 °C for 10 s, annealing at 59 °C for 20 s, and elongation at 72 °C for 15 s. PCR products were visualised on the QIAxcel System (QIAGEN GmbH, Hilden, Germany) using the QIAxcel DNA High Resolution Kit (QIAGEN GmbH, Hilden, Germany) with a QX Alignment Marker 15 bp/3 kb (QIAGEN GmbH, Hilden, Germany), the QX DNA Size Marker 100 bp–2.5 kb (QIAGEN GmbH, Hilden, Germany) and OM500 settings. The *tul4* fragments were sequenced using the forward and reverse primer on the 3130XL Genetic Analyzer using the real-time PCR products as template (Table 2).

### 2.3. Spectra Generation of Strains Used in this Study

For each strain (Table 1), spectra were generated using the direct transfer and the formic acid extraction method using HCCA (α-cyano-4-hydroxycinnamic acid in trifluoracetic acid) matrix (Bruker GMBH, Bremen, Germany) as described by the manufacturer. FlexControl version 3.3 (Bruker Daltonics GmbH, Bremen, Germany) was used to create spectra of the selected strains and perform baseline subtraction and smoothening of the spectra.

All *F. tularensis* strains were handled in a BSL-3 facility. Unviability of the organisms smeared on the MALDI target was confirmed by swapping two spots of each strain and leaving two spots for MALDI-TOF MS. The swabs were checked for growth on chocolate agar with vitox (PO5090A, Oxoid Deutschland GmbH, Wesel, Germany) aerobically (7% CO_2_) at 37 °C for 48 h before the MALDI target left the BSL-3 facility. Unviability of the organisms using the extraction method was confirmed by plating 25 µL of the milliQ/ethanol mixture on chocolate agar with vitox as described for the swabs.

### 2.4. Creation of Main Spectra (MSPs) from Reference Strains and Construction of In-House Francisella Library

To construct MSPs, the type strain of each *Francisella* (sub)species was used as a reference strain. However, for the avirulent *F. tularensis* ssp. *tularensis* type strain [3] and the not publicly available type strain of *F. tularensis* ssp. *holarctica* [26], *F. tularensis* ssp. *tularensis* Schu4 and *F. tularensis* ssp. *holarctica* LVS were used as reference strains, respectively (Table 1). A protein extract of each strain was applied to the target plate eightfold and analysed in triplicate, resulting in 24 spectra. MALDI Biotyper version 3.0 (Bruker Daltonics GmbH, Bremen, Germany) was applied to assemble the 24 spectra to an MSP and create an in-house *Francisella* library.

### 2.5. Verification of the Method of MSP Creation

To verify the method of MSP creation, the MSPs of *F. philomiragia* strains F184, F185, F93, and F51 from the MALDI Biotyper library version 3.1.2–version 11 (BDAL; Bruker Daltonics GmbH, Bremen, Germany) and the MSPs of *F. tularensis* ssp. *holarctica* strain 10,857 (=NCTC 10857) and *F. tularensis* ssp. *tularensis* strain CAPM 5600 (=Schu4) of the MALDI Biotyper Security library version 1.0.0.0 (SR; Bruker Daltonics GmbH, Bremen, Germany) were compared to in-house MSPs of the same strains by creating a dendrogram with an arbitrary distance normalised to a maximum value of 1000 (Bruker MALDI Biotyper Compass Explorer 4.1).

### 2.6. Identification of Francisella (sub)Species Using Commercially Available and In-House Libraries

All spectra were verified against the BDAL library version 3.1.2 and to the BDAL+SR library (SR library version 1.0.0.0). Additionally, the in-house *Francisella* library was verified using those strains. Verification was performed both with the direct transfer method and the formic acid extraction method on a Microflex LT mass spectrometer (Bruker Daltonics GmbH, Bremen, Germany) in linear positive mode measuring in a range of 2000–20,000 kDa [27]. The SR library does not discriminate between the subspecies; however, this database includes strain (CAPM) 5600, which is *F. tularensis* ssp. *tularensis*, and strains (CAPM) 5151, 5536, 5537, 5540 and (NCTC) 10,857 that are *F. tularensis* ssp. *holarctica*.

According to the manufacturer, scores between 2.300 and 3.000 are highly probable species identifications, scores between 2.000 and 2.299 are secure genus identifications and probable species identifications, scores between 1.700 and 1.999 are probable genus identifications, and scores below 1.700 are considered “no reliable identifications”. Because the subspecies identification of *F. tularensis* has a clinical impact, not only the differences in score between the first and second species (not to be confused with the score of the second hit), but also the differences in scores between subspecies will be determined. Differences in scores between duplicates of the same strain higher than 0.300 were rejected.

### 2.7. Biomarker Selection and Verification

For the selection of biomarkers suitable for discrimination of *Francisellaceae*, the MSPs of the reference strain of all species and subspecies of *Francisella* (Table 1) were imported as a text file from flexAnalysis into BioNumerics 7.1 (Applied Maths, Gent, Belgium) using the “preprocessing (strict)” method. Biomarkers were identified with the “peak class matching” method according to the standard settings except for “constant tolerance = 2” and “linear tolerance = 500”.

The selected biomarkers were validated using the spectra of all strains (Table 1) after both the direct transfer method and the formic acid extraction method. Peaks were considered to be (sub)species-specific biomarkers when they were present in all spectra of all strains of the specific species or subspecies and absent in all spectra of most other strains. Biomarkers were compared with the MSP profiles and should be present in all individual spectra of an MSP. Extra biomarkers related in mass or DNA sequence were added. *m*/*z* values of the selected biomarkers were also compared in FlexAnalysis with the mass values of the identification runs, i.e., the unexpected absence or presence of masses were examined manually on peak level because biomarkers can be missed, because routine analyses are based on less spectra than MSPs.

### 2.8. Identification of Proteins Related to Biomarkers

To identify potential proteins, the *m*/*z* value of the biomarkers were compared to the Tagident and to the UniprotKB database [28]. This was performed with 1+, 2+, and 3+ charges, as multiple charges have been observed using the HCCA matrix [29] and for the common post-translational maturations, methylation (+15, 30 and 45 Da), acetylation (+42 Da), and methionine loss at the N-terminus of the protein (−131 Da). Incidentally, proteins start with leucine, a functional start amino acid (AA) [30], and not the assigned methionine. Therefore, the molecular weight of the biomarkers was determined from the DNA sequence of the genes that was translated into amino acid by CloneManager. Because N-formylmethionyl-tRNA deformylase was present in the genomes of the reference strains (except CP010427), no correction was applied for the formyl mass.

### 2.9. Calculation of Biomarker Masses from DNA Sequences

Knowing the identity of the proteins, and as such their nucleotide sequence, made it possible to determine their presence in the relevant (sub)species and absence in the other (sub)species using the BLASTn algorithm in the NCBI Nuleotide Collection (nr/nt) database April–May 2022 against “*Francisellaceae*”. The results were downloaded as a seqdump.txt file and aligned and translated to an AA sequence with CloneManager for comparison with the predicted AA sequence of the reference strains. The mass of the AA sequence was calculated with CloneManager (designated theoretical *m*/*z* value throughout this manuscript). For the 50S ribosomal protein L34, a BLASTp was performed to check its genus specificity. Manual adjustments were made when necessary. *F.* salinarina [19], *F. oppertunistica* [31], *A. frigiaqua*, *A. inopinata* [32], and *Pseudofrancisella aestuarii* [33] are genetically related to *F. tularensis*. Except for *F. oppertunistica*, the species were found in cold (sea)water without any relation to human pathogenicity. However, to avoid misidentification, the theoretical presence of the biomarkers was also determined for these species.

## 3. Results

### 3.1. Verification of the Strain Identity

The identity of all strains included in this study was confirmed using 16S rDNA Sanger sequencing, detection of the *tul4* and *FT-heli* genes (Table 1), and biochemical methods. One strain of *F. tularensis* ssp. *novicida* was negative in the real-time PCR but demonstrated a positive *tul4* band on agarose gel. Sequencing of this product revealed a discrepancy of two bases compared to the Fl-probe sequence. Of the genetically and phenotypically related species, *tul4* was only detected in *F. hispaniensis*.

The FT-heli target was positive in all *F. tularensis* strains and the size of the target differed between *F. tularensis* ssp. *holarctica* (153 bp) and the other *F. tularensis* subspecies (183 bp) as described [22]. However, this last target was not specific for *F. tularensis* because FT-heli was also detected in *F. hispaniensis* and *F. philomiragia* and both displayed a 183 bp PCR product.

### 3.2. Spectra Generation of Strains Used in this Study

The application of standard methods for direct transfer and formic acid extraction left no viable *F. tularensis* cells on the MALDI target plate, confirming that transport outside of the BSL3-facility is safe.

### 3.3. Verification of the Method of MSP Creation

The MSPs of both the Bruker databases and the in-house database formed separated clusters per species (Figure 2), indicating differences on species level that are possibly suitable for biomarker-based identification. The SR MSPs of strains 5600 (Schu) and 10,857 did not cluster directly with the in-house MSP of the respective strains of *F. tularensis* ssp. *tularensis* BD13-00126 and *F. tularensis* ssp. *holarctica* BD13-00127. The four corresponding *F. philomiragia* strains F51, F93, F184, and F185 formed database-related clusters, indicating differences in preparing MSPs.

### 3.4. Identification of Francisella (sub)Species Using Commercially Available and In-House Libraries

Using only the BDAL library, all *Francisella* strains demonstrated “no reliable identification”. This is a correct score as none of these *Francisella* species are present in this database. *F. hispaniensis* was incorrectly identified as *Lactobacillus diolivorans* with a borderline log score of 1.760, indicating “probable genus identification” when using the formic acid extraction method. *F. philomiragia* and *Legionella pneumophila* were correctly identified.

Using the BDAL+SR library, all *F. tularensis* subspecies were correctly identified as *F. tularensis* (Appendix A), but no subspecies differentiation is possible with these libraries. The subspecies *tularensis* and *holarctica* could however be differentiated by their strain number (Table 3) with both direct transfer and formic acid extraction method. The *F. tularensis* ssp. *tularensis* strains all scored “*F. tularensis* 5600” above 2.000 and the difference with the second species was at least 0.221. *F. tularensis* ssp. *holarctica* and *F. philomiragia* strains were correctly identified with scores higher than 1.800, and without second species scores. The strains of the other two *F. tularensis* subspecies and *F. hispaniensis* scored *F. tularensis* below 2.000 without discriminating between subspecies *tularensis* and *holarctica*.

The in-house *Francisella* library can detect *F. tularensis* down to a subspecies level (Figure 3) with log scores higher than 2.000 and sufficient differences with the second species. However, *F. tularensis* ssp. *mediasiatica* and *F. tularensis* ssp. *novicida* incidentally scored lower than 2.000 using the direct transfer method (>1.900, see Appendix A). In addition, the other *Francisella* species were identified correctly at species level with log scores higher than 2.500 when using the extraction method. *L. pneumophila* was correctly identified as “no reliable ID” with the in-house *Francisella* library, because this species was not included.

### 3.5. Biomarker Selection and Verification

Biomarker identification by BioNumerics from the MSPs of the reference strain of all *Francisella* species and subspecies resulted in the selection of 17 biomarker masses that were potentially suitable for biomarker-based identification (Appendix A). Verification of the selected biomarkers using the spectra of all strains included in the study (Table 1) warranted the additional inclusion of masses 2195 and 5182–5185, resulting in a total of 19 selected biomarkers (Table 4). The 2195 Da marker was not present in the corresponding MSP of *F. tularensis* ssp. *mediasiatica*, probably because it was missing in one of the twenty-four spectra used for building the MSP. The 5182–5185 mass was excluded by BioNumerics as not being (sub)species specific. It is however included as a *Francisella* genus-specific marker because it was identified as such earlier [13,14,15,16,17,18].

Biomarker selection was further investigated by a manual check of their presence in the MSPs, their presence in the peak lists of all strains, and the theoretical *m*/*z* value of the matching protein calculated from the DNA sequence of the reference strain (Table 4). This resulted in the addition of six *m*/*z* values to the selected biomarkers. The *m*/*z* value 3130–3132 of the hypothetical protein was detected in the MSP of the *F. philomiragia*. A peak with *m*/*z* value 3119 was observed in de spectra of the *A. guangzhouensis* but was not observed in the MSP of the *A. guangzhouensis*. *m*/*z* values 5113, 5187, 9360, and 9420–9423 were also added because of close relatedness in either mass or DNA sequence.

### 3.6. Identification of Proteins Related to Biomarkers

The validated biomarkers were matched with possible proteins in the genome sequence of the respective reference strains (Appendix A). The selected biomarkers could all be linked to a single protein with the exception of the *m*/*z* values 3119 and 4040 that could not be assigned to a gene. The selected biomarker 3679 Da (Appendix A) was specific for *F. orientalis* was assigned to several related cold shock proteins, but BLASTn analyses indicated that this mass is less suitable as a biomarker, because similar proteins with comparable weight are found in the spectra of four (sub)species and on the genomes of *A. frigidaqua* and *P. aetuarii* (Appendix A). Therefore, this biomarker was excluded from the selection of suitable biomarkers (Table 4). Verification of the selected biomarker with *m*/*z* value 4387 (Appendix A) demonstrated that a peak was also present in the spectra of *F. tularensis* ssp. *novicida*. This peak could not be assigned to BolA. However, a *Bol A* gene coding for a protein with similar molecular weight was present in *A. innopinata* and *P. aetuarii*. Therefore, this biomarker was also excluded from Table 4 as a suitable biomarker.

### 3.7. Francisella Identification by Biomarker

The scheme provided in Table 4 can be used for the identification of *Francisellaceae*. According to our data and the literature [13,14,15,16,17,18,19], the 5182–5185 mass of the 50S ribosomal protein L34 was specific for the *Francisella* genus. Within the *Francisella* genus, the species *F. tularensis* could be differentiated from the other *Francisella* species by the histone-like protein form Beta (Hu-beta) mass (selected biomarker 9449). The 9444 Da variant of Hu-beta was observed in the genome of subspecies *novicida*. Although the 9449 Da mass was found in the MSPs of the other subspecies, a corresponding *Hu-beta* gene was not present in any of investigated genomes of the subspecies *tularensis*, *holarctica*, or *mediasia-tica*. Instead, a *Hu-beta* variant with a theoretical mass of 9474 Da, caused by a single point mutation, was observed in their genome. This variant was found in only one *F. tularensis* ssp. *novicida* strain, namely, TCH2015. The theoretical *m*/*z* value of 9444 is also present in *F. opportunistica* genome (Table 4) and in that of *F. persica* (Appendix A), although sufficient expression of the protein to determine a peak was not confirmed.

The subspecies differentiation for *F. tularensis* was possible with the 50S ribosomal protein L36 (*F. tularensis* ssp. *mediasiatica*; 2195 Da), the unknown mass of 4040 Da (*F. tularensis* ssp. *novicida*) and the 50S ribosomal protein L27 (*F. tularensis* ssp. *holarctica*; 4455 Da). Additionally, selected biomarker 8079 was observed in the MSP of *F. tularensis* ssp. *novicida* corresponding to a DNA-directed RNA polymerase Ω subunit (theoretical *m*/*z* value 8082). Even though this last biomarker was not observed in the MSPs and spectra of the other *F. tularensis* subspecies, the gene was present in their genome and also in that of *F. hispaniensis*.

To differentiate the *Francisella* species other than *F. tularensis*, both the previously mentioned Hu-beta protein and the 50S ribosomal proteins L27 and L29 can be used. The Hu-beta protein displayed different masses for *F. noatunensis* subspecies *noatunensis* (selected biomarker 9380), *F. hispaniensis* and *F. orientalis* (*m*/*z* value 9420–9423) and *F. philomiragia* (*m*/*z* value 9394–9396). The HU beta protein in the *F. salimarina*-genome had a theoretical mass of 9392 Da. Selected biomarker 3128 was also unique for *F. noatunensis* ssp. *noatunensis* and corresponded to a hypothetical protein with a theoretical mass of 3129 Da.

The 50S ribosomal protein L27 was observed as a peak of 4477 in the MSPs of *F. orientalis* and *L. pneumophila*. However, in the *F. philomiragia* spectrum a peak of 8923–8925 is observed, caused by the +2 charge of the same protein (Appendix A). This difference is based on a single point mutation and was also observed in the genome sequence of *F. salimarina*. Additionally, the 50S ribosomal protein L29 can be used to differentiate *F. philomiragia* from the other *Francisella* species. This protein is observed as 3880–3881 peak (+1 charge) and as 7759–7761 (+2 charge). The protein was also found on the genome of *F. oppertunistica* and could in theory be detected with both charges as well. The 50S ribosomal protein L33 is specific for *F. orientalis* (selected biomarker 3101) and a hypothetical protein (selected biomarker 3130–3132 Da) is only present in *F. philomiragia*.

The *F. noatunensis* ssp. *noatunensis* biomarker of 10,215 Da matched with two different genes: the Chaporin protein *GroES* and the Parvulin-like peptidyprolyl isomerase. This Parvulin-like peptidyprolyl isomerase gene was also present in the genomes of *F. oppertunistica* and *F. tularensis* ssp. *novicida*, although a corresponding peak was not observed in the MSP or in the spectra of the latter subspecies.

*F. orientalis* could be identified using selected biomarkers 3101 and 4477. Although the latter was also observed in *L. pneumophila*, *m*/*z* 3101 is absent in this species.

The *Allofrancisella* species could be identified using selected biomarkers 5113 and 9360, which corresponded with the 50S ribosomal protein L34 and the Hu-beta protein, respectively. Both corresponding genes were also found in *A. frigidaquae* and *A. innopinata* (with theoretical masses of 5110 and 9357–9359, respectively), making it a suitable biomarker for the *Allofrancisellaea*.

An unknown protein of 3119 Da and one of 5187 Da were found in the MSPs of *A. guangzhouensis* and might be species specific. However, their usefulness in differentiation from the other *Allofrancisella* species should still be confirmed in vitro.

From the genome of *Pseudofrancisella aetuarii* a unique mass of 5260 Da could be identified for the 50S ribosomal protein L34, although in vitro confirmation is needed. Additionally, a specific *Hu-beta* gene with a calculated molecular weight of 9673 Da was found in the *P. aetuarii* genome.

## 4. Discussion

MALDI-TOF MS is increasingly used in clinical laboratories for the identification of bacteria. In this study, the application of the Bruker Biotyper system was assessed as a fast and specific method for the detection and identification of *F. tularensis* to the subspecies level in a clinical laboratory setting. Adding subspecies differentiation directly to this method will improve speed and certainty under which safety conditions to work.

Because MSPs were both used as basis for the in-house *Francisella* library and the selection of biomarkers, the method of MSP creation was verified as shown in the dendrogram of the MSPs. This demonstrated sufficient differences between species, but the spectra as provided by Brucker did not cluster with in-house generated spectra of the same strains as already shown by Seibold et al. [13]. The *F. tularensis* ssp. *tularensis* strain 5600 clusters with *F. tularensis* ssp. *holarctica* strains, whereas our in-house generated spectrum of this same strain (BD13-00126) clustered with *F. tularensis* ssp. *mediasiatica*. There was no indication for differences in the quality of the spectra. Because the same extraction procedure and matrix were used in the Bruker databases and the in-house database, growth conditions are the most probable cause for discrepancies. Particularly, *F. philomiragia* was cultured on chocolate agar with vitox in this study, whereas HCA agar was used for developing the Bruker databases. The effect of the lack of standardization of culture media was recently reviewed [34] and indicated that identification should be performed under the exact same (growth) conditions as those under which the database was built as was performed with the in-house database and the strains tested.

*F. tularensis* is absent in the standard libraries of the Bruker Biotyper System and Vitek MS. *F. tularensis* is included in the MALDI Biotyper Security library for highly pathogenic (BSL-3) bacteria, however without the possibility of *F. tularensis* subspecies differentiation. In this study, we demonstrated that *F. tularensis* ssp. *tularensis* and *holarctica* can be identified with the SR library using both the direct transfer method and the formic acid extraction method. Care should be taken with scores below 2.300, because *F. hispaniensis* and the *F. tularensis* subspecies *novicida* and *mediasiatica* scored *F. tularensis* ssp. *tularensis* or *holarctica* in the range 1.760–2.200. Even though the human pathogen *F. hispaniensis* causes severe illness [4] and is genetically more closely related to *F. tularensis* than to other *Francisella* species, *F. hispaniensis* misidentification as *F. tularensis* may cause unnecessary alarm. Unlike *F. tularensis*, *F. hispaniensis* is not associated with intentional release. Identifying *F. tularensis* with the MALDI Biotyper Security library should be considered as a warning to directly verify the subspecies level using the biomarkers and when necessary, proceed handling of the bacterium under BSL-3 conditions.

Not all laboratories using the Bruker Biotyper system have obtained the SR library because of the extra costs and tedious administration. Analysis of a *Francisella tularensis* isolate using only the MALDI Biotyper library will not give false positive results with other, less virulent, organisms. However, using only this library will not give a warning for the presence of *F. tularensis* either. With the Vitek MS, *F. tularensis* cannot be detected because the library does not contain any *Francisellaceae* [12]. To overcome the above problems, an in-house *Francisella* library is an excellent alternative. The spectra are sufficiently distinctive to distinguish to *Francisellaceae* at (sub)species level as previously determined for *F. tularensis* ssp. *tularensis* and *F. tularensis* ssp. *holarctica* [15]. Spectra generated as part of the in house developed library can be exchanged with other laboratories.

The use of biomarkers excludes the need for acquiring an external database or implementing an in-house *Francisella* library for which highly pathogenic strains must be obtained and cultivated. Moreover, the Bruker MALDI-TOF MS has the option to create libraries; however, other MALDI-TOF MS systems such as the Vitek MS lack this possibility. The described biomarkers could potentially be useful in Vitek-MS analyses as well; however this requires thorough validation.

In this study, biomarkers were identified for all *F. tularensis* subspecies and the other *Francisella* species irrespective of the extraction method used. The fact that most of the selected biomarkers are highly expressed conserved proteins makes them suitable for identification of strains grown under different conditions [33]. However, various genes coding for proteins related to specific biomarkers were present in several species but were not expressed or insufficiently expressed to reach the peak threshold under the conditions used in this setup. This indicates that under different growth conditions those peaks may be found, and specificity cannot be based on those peaks only. Additionally, masses translated from DNA do not always lead to the observation of a shifted peak in the MALDI-TOF spectrum. Therefore, a combination of multiple biomarkers is necessary for an accurate identification. Of the peaks found in this study, the Hu-beta peak was described previously [13,14]. Other peaks described previously were not found either because they were not differentiating within the study [13] or due to the limited number of (sub)species tested [14]. However, in our study we focused on smaller peaks to find peaks that were not previously included as potential biomarkers.

Combining biomarker selection with identification of the differentiating peaks makes this technique suitable to use on whole genome sequences of clinical strains to both improve the MALDI-TOF MS identification and the identification of whole genome sequences. In samples where culturing proves difficult, this could be an additional method for identification.

## 5. Conclusions

Standard MALDI-TOF MS libraries do not (yet) contain *F. tularensis* MSPs. The additional Bruker MALDI Biotyper Security library can identify *F. tularensis* ssp. *tularensis* and *holarctica* when strain identifiers are considered. An in-house *Francisella* library can be used effectively to determine the specific species of *Francisella* and subspecies of *F. tularensis*. Biomarkers can confirm the same after initial identification of *Francisella*, but without the difficulty of developing an in-house library containing spectra of bioterrorism agents or dependency on an external library. In conclusion, MALDI-TOF MS using biomarkers or a *Francisella* library can successfully be applied in a clinical laboratory setting as a fast and specific method to detect and determine *F. tularensis* subspecies, ensuring the safety of laboratory personnel.

## Figures and Tables

**Figure 1 microorganisms-11-00905-f001:**
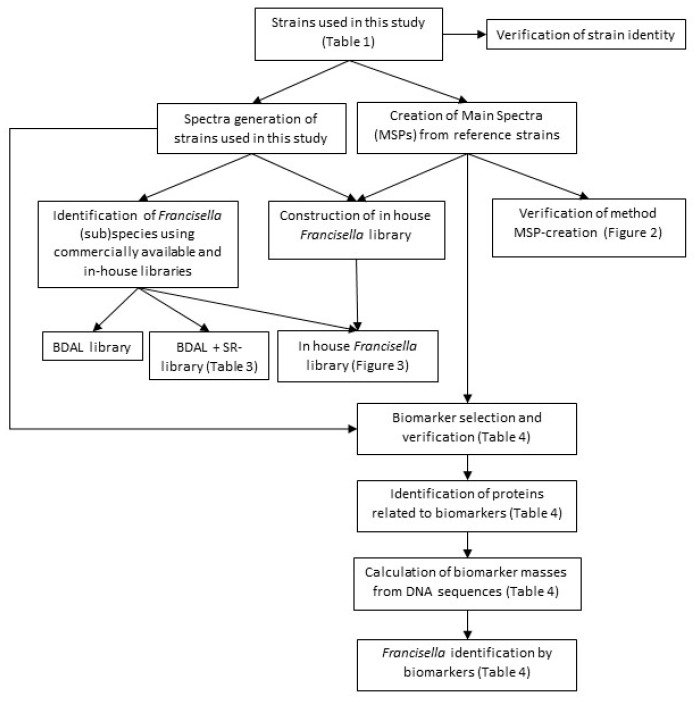
Flow chart of this study.

**Figure 2 microorganisms-11-00905-f002:**
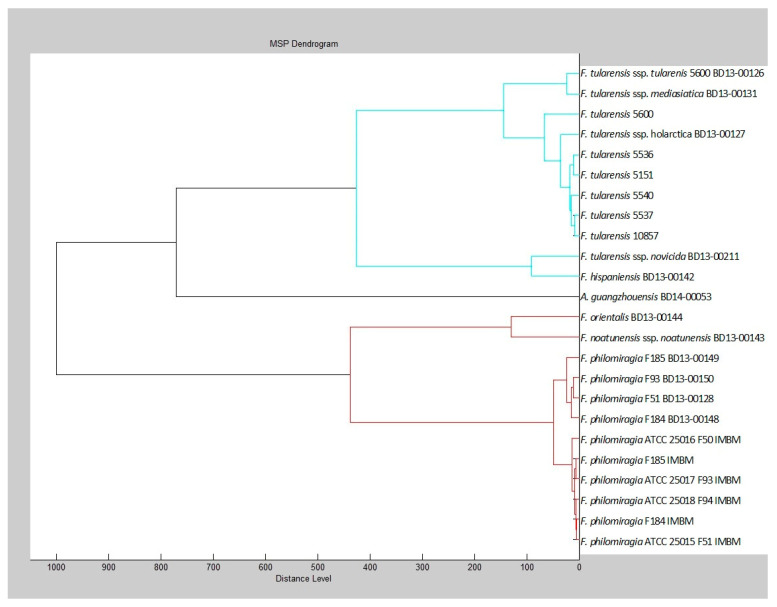
Dendrogram of the MSPs from the BDAL+SR library and MSPs from the in-house *Francisella* database: the maximal distance level is set on maximal 1000 units. The blue cluster represents *F. tularensis* and *F. hispaniensis* species; the strain numbers are from SR library. The red cluster represents the other *Francisella* species; species marked IMBM are from the BDAL library, all other species are from the in-house database.

**Figure 3 microorganisms-11-00905-f003:**
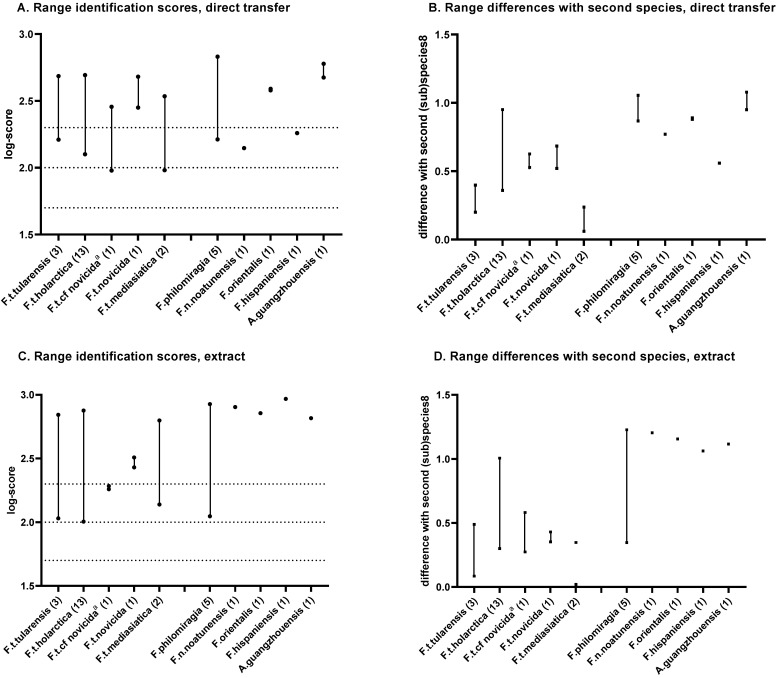
Identification log-scores of *Francisellaceae* strains with the in-house *Francisella* database. Identification log-score with direct transfer (**A**) and extraction methods (**C**); differences in log-scores with the next species scored for direct transfer (**B**) and extraction methods (**D**). The 1.7, 2.0, and 2.3 log-scores are indicated with dotted lines; ^a^ A score with *F. tularensis* ssp. *novicida* was considered correct; (): number of strains tested.

**Table 1 microorganisms-11-00905-t001:** Strains and molecular strain identification.

Species	Identification Number	Source Number	% homology with 16S rDNA Type Strain ^a^	Cp *tul4*	Cp FT-heli	Fragment Size FT-heli
** *Francisella* ** ** *tularensis* **						
*Francisella tularensis* ssp. *tularensis*	BD13-00202	Ft30 = ATCC 6223 T	100.0%	19.23	19.78	183 bp
	BD13-00126	Schu4 = CAPM 5600	99.9%	21.68	21.51	183 bp
	BD06-00286	03-01300	100.0%	19.00	19.68	183 bp
*Francisella tularensis* ssp. *holarctica*	BD06-00290	03-1292	100.0%	21.32	22.03	153 bp
	BD11-00177	Wild-type	100.0%	17.41	17.70	153 bp
	BD07-00537	Wild-type	100.0%	18.07	19.42	153 bp
	BD13-00130	FT32	100.0%	19.89	20.49	153 bp
	BD13-00127 ^b^	LVS = NCTC 10857	100.0%	20.60	21.11	153 bp
	BD13-00203	A104-15	100.0%	20.44	20.90	153 bp
	BD13-00204	A104-16	100.0%	19.06	19.70	153 bp
	BD13-00205	A101-6	100.0%	17.17	17.88	153 bp
	BD13-00206	A146-2	100.0%	18.72	19.26	153 bp
	BD13-00207	A155-4	100.0%	17.48	18.00	153 bp
	BD13-00208	A187-9	100.0%	17.54	18.52	153 bp
	BD13-00209	Ft 28	100.0%	18.92	19.64	153 bp
	BD15-00089	Wild-type	100.0%	14.47	14.78	153 bp
	BD16-00254 ^c^	Wild-type	100.0%	22.11	20.92	153 bp
	BD16-00255 ^c^	Wild-type	100.0%	21.54	20.20	153.bp
	BD18-00109 ^c^	Wild-type	100.0%	12.87	20.23	153.bp
	BD18-00117 ^c^	Wild-type	100.0%	13.32	20.90	153.bp
	BD20-00065 ^c^	Wild-type	99.8%	18.63	17.66	153.bp
	BD21-00021 ^c^	Wild-type	100.0%	12.10	19.17	153.bp
	BD21-00041 ^c^	Wild-type	100.0%	12.68	1957	153.bp
	BD21-00091 ^c^	Wild-type	100.0%	11.30	15.05	153 bp
*Francisella tularensis* ssp. *mediasiatica*	BD13-00131	Ft31	100.0%	20.81	21.09	183 bp
	BD13-00210	Ft 24 T = U112	100.0%	18.97	19.44	183 bp
*Francisella tularensis* ssp. cf *novicida*	BD13-00129	Ft26 = Fx1	99.9%	-	20.10	183 bp
*Francisella tularensis* ssp. *novicida*	BD13-00211	Ft 27 T = FSC147	100.0%	16.84	17.22	183 bp
**Genetically related species**						
*Francisella hispaniensis*	BD13-00142	DSMZ 22,475 T	99.9%	19.77	35.33	183 bp
*Francisella noatunensis* ssp. * noatunensis*	BD13-00143	DSMZ 18777	100.0%	-	-	-
*Francisella orientalis*	BD13-00144	DSMZ 21,254 T	100.0%	-	-	-
*Francisella philomiragia*	BD13-00148	CCUG 12,603 = F184	100.0%	-	25.85	183 bp
*Francisella philomiragia*	BD13-00149	CCUG 13,404 = F185	100.0%	-	24.92	183 bp
*Francisella philomiragia*	BD13-00150	CCUG 19701 = ATCC25017 = F93	100.0%	-	26.18	183 bp
*Francisella philomiragia*	BD13-00128	DSMZ 7535 T = ATCC25015 = F51	100.0%	-	21.71	183 bp
*Allofrancisella guangzhouensis*	BD14-00053	CCUG 60,119 T	100.0%	-	-	-
**Phenotypically related species**						
*Legionella* *pneumophila*	BD99-00039	ATCC 33,152 T	100.0%	-	-	-

Strains included in the in-house *Francisella* library and biomarker selection are shaded; ^a^ T (type strain) 16S rDNA sequence compared to in-house sequence of the type strain; ^b^ A QIAxcel band was present; sequencing demonstrated two base pairs difference with the fluorescent-labelled probe; ^c^ Not included in the validation of *Francisella* (sub)species using commercially available and in-house libraries), but identification was applied in the routine setting with de in-house database.

**Table 2 microorganisms-11-00905-t002:** Primers and probes.

Target	Primer/Probe	Sequence (5’-3’)
*16S*	16S8-forward	AGA GTT TGA TCM TGG YTC AG
	16S575-reverse	CTT TAC GCC CAR TRA WTC CG
*tul4*	forward primer	CGC GCG GAT AAT TTA AAT TTC TC
	reverse primer	CAT ACT GTT GGA TAG GTG TTG GA
	fluorescein-labelled probe	CTA CAA TAT CTT GAT TCA GCC CAA GCT G **FL**
	CAL Fluor Red 635-labelled probe	**CF-red** CTA AAA TCT TTT TTA GTT TCA GAA TTC ATT TTT GTC CGT
*FT-heli*	forward primer	CTT TAT CAA TCG CAG GTT TAG C
	reverse primer	GGA AGC TTG TAT CAT GGC ACT
	fluorescein-labelled probe	ACA GCA TAC AAT AAT AAC CCA CAAG GAA GT **FL**
	CAL Fluor Red 635-labelled probe	**CF-red** TAA GAT TAC AAT GGC AGG CTC CAG AA

**Table 3 microorganisms-11-00905-t003:** *Francisella* identification scores against BDAL+SR databases.

	*n*	Direct Transfer		Extract	
		Range Correct Score	Difference with Second (sub)Species ^a^	Range Correct Score	Difference with Second (sub)Species
*F. tularensis* ssp. *tularensis*	3	2.194–2.478	0.250–0.410	2.023–2.449	0.221–0.378
*F. tularensis* ssp. *holarctica*	13	1.927–2.484	no second species	2.080–2.381	0.300–0.447 ^b^
*F. philomiragia*	5	1.919–2.355	no second species	1.841–2.425	no second species
*F. noatunensis* ssp. *noatunensis*	1	No ID		No ID	
*F. orientalis*	1	No ID		No ID	
*A. guangzhouensis*	1	No ID		No ID	
*F. hispaniensis*	1	Not applicable		No ID	
		**Range incorrect score**		**Range incorrect score**	
*F. tularensis* cf *novicida*	1	1.782–1.880	no second species	1.817–1.958	
*F. tularensis* ssp. *novicida*	1	1.862–1.986	no second species	1.894–2.002	
*F. tularensis* ssp. *mediasiatica*	2	1.860–2.200	no second species	1.812–2.097	
*F. hispaniensis*	1	1.760	no second species	Not applicable	

The BDAL library contains only *F. philomiragia*, the SR library contains only *F. tularensis* ssp. *tularensis* and *F. tularensis* ssp. *holarctica*, but the manufacturer does not differentiate between them. In this table, the subspecies were differentiated *F. tularensis* ssp. *tularensis* = strain 5600; *F. tularensis* ssp. *holarctica* = strain 5151, 5536, 5537, 5540, or 10,857 (see also Appendix A); no match with the correct (subs)species was considered an incorrect score. n = number of strains; second species scores always < 2.000 unless stated otherwise; ^a^ when the second species scored < 1.7, the difference was calculated from 1.7; ^b^ only for 1 out of 13 strain the second species scored > 0.2000.

**Table 4 microorganisms-11-00905-t004:** *Franciscellaceae* Identification by biomarker *m*/*z* values.

Selected biomarkers *m*/*z*				2195	3101	3128	3130–3132		3880–3881	4040	4455	4477	5113	5187	5182–5185		7759–7761	8079	8522	8923–8925	9360	9380	9394–9396	9420–9423	9449	9476–9477		10,215
Observed *m*/*z* in DS/Extract				2191–2196	3096–3099	3123–3127	3129–3131	3119	3878–3881	4035–4039	4450–4455	4473–4476	5113	5187	5175–5185		7756–7761	8071–8076	8514–8521	8920–8926	9360	9370–9375	9390–9396	9411–9421	9440–9446	9469–9480		10,206–10,210
Matching protein				50S-L36	50S-L33	Hyp. protein	Hyp. protein	UNK	50S-L29	UNK	50S-L27	50S-L27	50S-L34	UNK	50S-L34	50S-L34	50S-L29	RNA polym Ω	PAP2	50S-L27	Hu beta	Hu beta	Hu beta	Hu beta	Hu beta	Hu beta	Hu beta	GroES/PPiso
Theoretical *m*/*z*				2197	3101	3129	3173 + 15	?	3879	UNK	4452	4475	5110	UNK	5180–5181	5260	7757	8082	8521	8922	9357–9359	9378	9392	9417	9444	9474	9673	10,212–10,214
Genus	Species	Subspecies	Source																									
*Francisella*	*F. tularensis*	*tularensis*	H															T										
		*holarctica*	H								PM							T *	T									
		*mediasiatica*	H															T										
		*novicida*	H																T							PM		T
	*F. hispaniensis*		H															T										
	*F. noatunensis*	*noatunesis*	F																									
	*F. orientalis*		F																									
	*F. philomiragia*		H																									
	*F. salimarina ^G & L^*		SW																									
	*F. opportunistica ^G^*		H																									
*Allofrancisella*	*A. guangzhouensis*		W	UNK																								
	*A. frigidaquae ^G^*		W																									
	*A. innopinata ^G^*		W																									
*Ps.Fr.*	*P. aetuarii ^G^*		SW																									
*Legionella*	*L. pneumophila*		H/W																									

Presence of a peak is indicated in dark grey. Theoretical presence of biomarker in strains not tested are indicated in light grey; ^L^ = MALDI-TOF MS data from reference Li et al.; ^G^ = only genome-based analyses; H: Human, F: fish, SW: seawater, W: water; T = theoretical mass, but no peak observed; PM = point mutation; UNK = unknown; *F.* noat. = *F. noatunensis*, Ps.fr. = *Pseudofrancisella*, Legi = *Legionella*; * Strain BD11-00177 *m*/*z* 8072.

## Data Availability

The data presented in this study are available in the Appendix A.

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
