# Peer review of "Identification of Francisella tularensis Subspecies in a Clinical Setting Using MALDI-TOF MS: An In-House Francisella Library and Biomarkers"

_microorganisms, 2023, doi:10.3390/microorganisms11040905_

Round 1

Reviewer 1 Report

Comments and Suggestions for Authors

Dear authors,

please explain better the Francisella DNA extraction procedure (indicate extraction kit) in paragraph "2.2 Checking the identity of the strain". The Francisella strains were inativacted in BSL-3 facility and then extrated in a BLS-2 laboratory?

Line 107: For the sequencing do you use 16S rDNA primers? Please indicate the primers sequences in Tabella 2, reference and PCR conditions or only the reference if primer and PCR conditions are decribed in the article.

Line 108 and line 130: For the real time PCR for the targeting of tul4 gene you indicated reference 21, but this article is not available online, please check if the reference is correct.

Line 144: In Table 2 please indicate in column "sequence"  the sense of the strand (5'-3') and check that the primers are write in the correct sense. Add also the sequence of 16S rDNA primers.

Regards

Author Response

Dear Reviewer,

Thank you for your very useful comments. We have adapted the manuscript as described below.

Please explain better the Francisella DNA extraction procedure (indicate extraction kit) in paragraph "2.2 Checking the identity of the strain". The Francisella strains were inativacted in BSL-3 facility and then extrated in a BLS-2 laboratory?

The sentence “A loop full of bacterial culture was suspended in 100 µl of sterile water and heat in-activation at 100°C for 30 min. under bsl3 conditions. All further molecular techniques were applied at bsl2-conditions.” is added to describe the way the DNA is obtained and under which conditions (lines112-114).

Line 107: For the sequencing do you use 16S rDNA primers? Please indicate the primers sequences in Tabella 2, reference and PCR conditions or only the reference if primer and PCR conditions are decribed in the article.

The sentence “PCR-amplification of the DNA consists of 10 min pre-denaturation at 95°C; 35 cycles of denaturation at 95°C for 30 s, annealing at 54°C for 30 s, and elongation at 72°C for 80 s using the primers described in table 2 [24].” is added (lines 114-116).

Line 108 and line 130: For the real time PCR for the targeting of tul4 gene you indicated reference 21, but this article is not available online, please check if the reference is correct.

Unfortunately the reference 21 is indeed not available online. To avoid uncertainties in the protocol the primers were added in table 2 of this paper.

Line 144: In Table 2 please indicate in column "sequence"  the sense of the strand (5'-3') and check that the primers are write in the correct sense. Add also the sequence of 16S rDNA primers.

The sequence of the 16S primers are provided in Table 2 and the direction of the primers and probes is added to the header.

With kind regards,

Dr. Maaike de Vries

Reviewer 2 Report

Dear Author, I reviewed the manuscript (microorganisms-2300765) entitled Identification of Francisella tularensis subspecies in a clinical setting using MALDI-TOF MS: An in-house Francisella library and biomarkers. This manuscript presents relevant information about F. tularensis identification. However, some sections of the submitted data can be improved. For this reason, I consider that this manuscript needs minor changes to be considered for publication in this journal. 

Additional comments.

Highlight the advantages of using MALDI-TOF MS to identify bacteria subspecies.

Check paragraph extension in this manuscript.

Compare the obtained findings with similar assays where MALDI-TOF MS identified similar bacteria. 

Include future trends to keep working with the obtained data. 

Try to conclude with a general statement of the most relevant part of this study.

Author Response

Dear Reviewer,

Thank you for your very useful comments. We have adapted the manuscript as described below.

Highlight the advantages of using MALDI-TOF MS to identify bacteria subspecies.

"Adding subspecies differentiation directly to this method will improve speed and certainty under which safety conditions to work" is added (lines 445-446).

Check paragraph extension in this manuscript.

This is checked and adapted at line 182, 259, 371, 377 and 481.

Compare the obtained findings with similar assays where MALDI-TOF MS identified similar bacteria. 

“as previously determined for F. tularensis ssp. tularensis and F. tularensis ssp. holarctica [15]”was added to the sentence “The spectra are sufficiently distinctive to distinguish to Francisellaceae at (sub)species level”. (lines 484-485)

Additionally the following part was added: “Of the peaks found in this study, the Hu-beta peak was described previously [13, 14]. Other peaks described previously were not found either because they were not differentiating within the study [13] or due to the limited number of (sub)species tested [14]. However, in our study we focused on smaller peaks to find peaks that were not included as potential biomarkers previously.” (lines 503-507)

Include future trends to keep working with the obtained data. 

The following paragraph was added: “Combining biomarker selection with identification of the differentiating peaks makes this technique suitable to use on whole genome sequences of clinical strains to both improve the MALDITOF MS identification and the identification of whole genome sequences. In samples where culturing proves difficult, this could be an additional method for identification.” (lines 508-512)

Try to conclude with a general statement of the most relevant part of this study.

The last sentence is transformed to “Concluding, MALDI-TOF MS using biomarkers or a Francisella library can successfully be applied in a clinical laboratory setting as a fast and specific method to detect and determine F. tularensis subspecies ensuring the safety of laboratory personnel.” (lines 520-523)

With kind regards,

Dr. Maaike de Vries